# Beyond the Edge: Markerless Pose Estimation of Speech Articulators from Ultrasound and Camera Images Using DeepLabCut

**DOI:** 10.3390/s22031133

**Published:** 2022-02-02

**Authors:** Alan Wrench, Jonathan Balch-Tomes

**Affiliations:** 1Clinical Audiology, Speech and Language Research Centre, Queen Margaret University, Musselburgh EH21 6UU, UK; 2Articulate Instruments Ltd., Musselburgh EH21 6UU, UK; jbalchtomes@articulateinstruments.com

**Keywords:** multimodal speech, lip reading, ultrasound tongue imaging, pose estimation, speech kinematics, keypoints, landmarks

## Abstract

Automatic feature extraction from images of speech articulators is currently achieved by detecting edges. Here, we investigate the use of pose estimation deep neural nets with transfer learning to perform markerless estimation of speech articulator keypoints using only a few hundred hand-labelled images as training input. Midsagittal ultrasound images of the tongue, jaw, and hyoid and camera images of the lips were hand-labelled with keypoints, trained using DeepLabCut and evaluated on unseen speakers and systems. Tongue surface contours interpolated from estimated and hand-labelled keypoints produced an average mean sum of distances (MSD) of 0.93, s.d. 0.46 mm, compared with 0.96, s.d. 0.39 mm, for two human labellers, and 2.3, s.d. 1.5 mm, for the best performing edge detection algorithm. A pilot set of simultaneous electromagnetic articulography (EMA) and ultrasound recordings demonstrated partial correlation among three physical sensor positions and the corresponding estimated keypoints and requires further investigation. The accuracy of the estimating lip aperture from a camera video was high, with a mean MSD of 0.70, s.d. 0.56 mm compared with 0.57, s.d. 0.48 mm for two human labellers. DeepLabCut was found to be a fast, accurate and fully automatic method of providing unique kinematic data for tongue, hyoid, jaw, and lips.

## 1. Introduction

In speech science, kinematic analysis of speech articulators is a key methodology in the quantification of speech production [1]. It can be used to relate movement to muscle activation and the timing of neural control signals. Biomechanical engineers can evaluate their models, sociophoneticians can quantify changes in articulatory gestures, clinical phoneticians can assess progress after intervention for speech disorders, and speech technologists can use the objective measures as input for silent speech recognition or lip-reading.

Electromagnetic articulography (EMA) is an important method for measuring the kinematics of speech articulators in 3D space. It has an advantage over image-based techniques because it generates movement coordinates of keypoints on articulators, such as tongue tip, blade and dorsum, lips, and jaw. It is the preferred technique for kinematic speech studies and, since the decommissioning of X-ray microbeam facilities, unique in providing intraoral keypoint data. Limitations on where the 2 mm × 3 mm electromagnetic sensors can be attached means that movement of the posterior tongue surface and hyoid cannot be monitored.

Ultrasound tongue imaging and camera video of the lips and face are instrumental techniques within the budget of most speech laboratories and have become popular as a source of articulatory speech data. They are non-invasive, convenient, and suitable for field work. Dynamic MRI of the vocal tract is another fast-evolving imaging technique with the important ability to image all the structures in the vocal tract although with significant disadvantages in cost, access, temporal/spatial resolution, unnatural recording conditions and severe acoustic noise. All of the aforementioned imaging techniques provide data, which must be postprocessed to extract measurable dynamic and static features of the vocal tract. Postprocessing of speech articulator image data has almost exclusively taken the form of edge detection or boundary segmentation. Accurate boundaries are useful for estimating vocal tract area functions but not so useful for measuring kinematics of articulators. It is also the case that edge detectors are sometimes fooled by imaging artefacts.

Recent advances in computer vision and machine learning offer an alternative approach, learning the mapping between an entire articulatory image and keypoints, labelled by experts, which need not be related to an edge or boundary. This paper investigates the potential of such pose estimation deep neural nets. We show that pose estimation can estimate the position and shape of articulatory structures within an image to an accuracy matching that of a human labeller. The movement of estimated keypoints partially correlates with that of EMA sensors. Further, more rigorous investigation is required to establish the limits of pose estimation in this regard.

### 1.1. State-of-the-Art in Ultrasound Tongue Contour Estimation

In order to determine which edge detection methods to compare with pose estimation we will review the state-of-the-art. Early attempts to extract a tongue surface contour from a midsagittal ultrasound image of the oral cavity were based on active contours (aka snakes) [2]. The most frequently referenced technique is *EdgeTrak* [3], where a spline with up to 100 control points is iteratively attracted to contiguous edge features in the image. The technique must be “seeded” with a contour close to the desired edge. To avoid the need to seed by hand every frame in a movie sequence, it is common to hand-label the first image and proceed through the movie by seeding each following frame with the estimated position of the contour in the preceding frame. This process leads to a tendency for the estimated contour to drift away from the tongue contour over time and become longer or shorter [4]. This approach is also, by design, bound to find an “edge” (continuous line where pixels are brighter above than below or vice versa). It cannot estimate the position of the tongue where there is no edge. *SLURP* [5] forms the most recent and successful development of the active contour approach. It incorporates a particle filter to generate multiple tongue configuration hypotheses. These hypotheses are used as seeding for the active contour to avoid the problem of drift. It also employs an active shape model, trained on a small number of tongue contour samples, for the purpose of constraining the shape and iteratively driving the snake optimization.

Machine learning was first used to estimate ultrasound tongue contours by Fasel and Berry [6]. They report a mean sum of distances (MSD, see Appendix B) accuracy of 0.7 ± 0.02 mm for their deep belief network, *AutoTrace*, which is remarkable given there were only 646 inputs to the network, meaning each image was resized to 19 × 34 pixels. The high accuracy score can be explained by the holdout method commonly used for testing network performance whereby a small percentage of images are randomly selected from the same dataset used for training and isolated for testing. Due to the slow rate of change of tongue movement with respect to sampling frequency, many test images are therefore almost identical to images seen in the training set. This ‘holdout’ method of testing produces accuracy scores that are not representative of how the estimation network would perform on data from unseen speakers and recording conditions. This was demonstrated by Fabre et al. [7] who showed their MSD accuracy of 1.9 mm diminished to 4.1 mm when no image frames from a test speaker were used in training, even when the recording conditions (ultrasound model, probe geometry, depth, field of view and contrast) were the same. Fasel and Berry [6] used a 20% holdout. Xu et al. [8] used a holdout of 8% of their hand-labelled frames reporting a MSD accuracy of 0.4 mm. More recent work by Mozzaffari [9] used a 5% holdout. Akgul and Aslan [10] used a 44% holdout and reported an MSD of 0.28 mm.

Many previous attempts to use deep networks for tongue contour estimation (*BowNet* [9], *MTracker* [11], and *DeepEdge* [12]) have adopted the U-net architecture [13] or U-net-like architecture (*IrisNet* [9]). U-Net is a convolutional neural network, developed for biomedical image segmentation with an architecture designed to work with a few thousand training images. This network classifies pixels with a probability of them belonging to a learned boundary. *TongueNet* [9] constitutes the only previous report on using a network for landmark feature identification as opposed to image segmentation. The authors indicate that about 10 keypoints along the tongue surface is optimal for accurate performance (see Table 1).

*AutoTrace* [6] was re-evaluated along with *TongueTrack* [14] and *SLURP* by Laporte [5]. The MSD accuracy scores are summarised in Table 1.

From the contour estimation algorithms listed in Table 1, *SLURP*, *DeepEdge,* and *MTracker* report the best performance with MSD values of 1.7, 1.4, and 1.4 mm, respectively. The authors of these methods also provide code that can be freely downloaded. These algorithms are therefore selected for further investigation and comparison with DeepLabCut pose estimation.

### 1.2. Lip Contour Estimation

Estimating lip contours from video of the face has a similar history to ultrasound tongue contour estimation. Early attempts used Snakes [16] and Active Shape Models [17]. Kaucic et al. [18] used Kalman filters to track the mucosal (inner) and vermillion (outer) borders of the lips. There is a need for lip feature extraction for the speechreading/lipreading application. Since 2011, with the development of convolutional neural networks (CNNs), this approach has dominated. However, the CNN lip feature encoders form part of a larger network for speech recognition and are embedded with no means to extract the lip features.

In the speech science field, the most often referenced technique, and one currently still in use for estimating lip contours for gestural speech research, is from a 1991 PhD by Lalouche [19]. This requires the participant’s lips to be coloured blue. All blue pixels are then extracted from the image by chroma key, and post-processing is carried out to estimate the mucosal and vermillion borders. In a similar approach, but without the requirement for blue lips, King and Ferragne [20] have used the *semanticseg* function in the MATLAB deep learning toolbox to extract lip boundaries and postprocessed by fitting an ellipse to the boundary shape to give an estimate of width and height of the vermillion border.

The Lalouche chroma key method cannot operate on greyscale images so evaluation of DeepLabCut is compared here only with hand-labelling.

## 2. Pose Estimation

Recent advances in computer vision and machine learning have dramatically improved the speed and accuracy of markerless body-pose estimation [21]. There are a growing number of freely available toolkits that apply deep neural networks to the estimation of human and animal limb and joint positions from 2D videos. These include DeepLabCut [22], DeepPoseKit [23], and SLEAP [24]. These software packages all use Google’s open-source TensorFlow platform to build and deploy convolutional deep neural network models. The DeepLabCut toolkit (DLC) [21,22,25] has a broad user base and has continuing support so was selected here for evaluation of pose estimation in the speech domain.

### DeepLabCut

Once installed, the Python-based DeepLabCut toolbox is run using a simple graphical user interface (GUI) requiring no programming skills. The GUI makes it easy for users to label keypoints, train the convolutional neural network, apply the resulting model to identify pixel coordinates of keypoints in images, and output them in a simple comma separated text file. The processes for training and estimating pose with DeepLabCut are outlined in Figure 1. Auxiliary tools, for visualizing and assessing the accuracy of the tracked keypoints are also available within the DLC graphical user interface. Deep learning approaches require very large amounts of labelled data for training. Large, labelled corpora are publicly available for classical problems, such as facial landmark detection and body pose estimation [26,27], but not for ultrasound tongue image contouring. It is not practical to hand-label tens of thousands of ultrasound images, but it is possible to leverage existing networks trained on large datasets in one domain, and transfer learning to a new domain using only a few hundred frames. DLC applies transfer learning from object recognition and human pose estimation. With only a small number of training images and a few hours of machine learning, the resultant network can perform to within human-level labelling accuracy [22]. Here, we evaluate that performance claim on the domains of ultrasound tongue imaging and lip camera imaging.

Deep net architectures designed for markerless pose estimations are typically composed of a backbone network (encoder), which functions as a feature extractor, integrated with body part detectors (decoders). DeepLabCut provides a choice of encoders (MobileNetV2 [28], ResNet [29], or EfficientNet [30]), all with weights pretrained on the ImageNet corpus that consists of 1.4 million images labelled according to the objects they contain. The body part detector algorithms are taken from a state-of-the art human pose estimation network called DeeperCut [31] from which it takes its name. DeeperCut is in turn trained on the Max-Planck-Institut für Informatik human pose dataset [27], consisting of 25,000 images containing over 40,000 people with annotated body joints. The “Lab” nomenclature references the ability of DeepLabCut to transfer learning from human pose estimation to other domains, such as animals or medical images, using only a few labelled images, making the process manageable for a single research laboratory.

Encoders are continuously being redesigned to improve the speed and accuracy of object recognition, and it has been shown that this improved encoder performance feeds directly through to improve pose estimation [25], particularly with respect to:Shorter training times.Less hand-labelled data required for training.Robustness on unseen data.

For each labelled keypoint, the decoder produces a corresponding output layer whose activity represents a probability score-map (aka heat-map). These score-maps represent the probability that a body part is at a particular pixel [31]. During training, a score-map is generated with unity probability for pixels within a ‘pos_dist_threshold’ (default = 17) pixel radius of the labelled keypoint pixel and zero elsewhere. Mathis et al. recommend a 17-pixel distance threshold after experimenting on different threshold values for a 1062 × 720-pixel resolution video input.

It is possible to use features of the score-maps such as the amplitude and width, or heuristics such as the continuity of body part trajectories, to identify images for which the decoder might make large errors. Images with insufficient automatic labelling performance that are identified in this way can then be manually labelled to increase the training set and iteratively improve the feature detectors.

DeepLabCut can use deep, residual networks, with either 50, 101, or 152 layers (ResNet). The optional MobileNetV2 is faster for both training and analysis and make analysis with CPU (as opposed to GPU) feasible. EfficientNet encoders are also available.

Labelling the training set with multiple related anatomical keypoints improves the accuracy of individual keypoint estimates. Mathis [22] shows a network, trained with all body part labels simultaneously, outperforms networks trained on only one or two body parts by nearly twofold. DeepLabCut applies image augmentation to artificially expand the training set by modifying the base set with images transformed by scaling, rotating, mirroring, contrast equalization, etc. In this paper, only scaling and rotation were applied.

## 3. Materials and Methods

### 3.1. Ultrasound Data Preparation

#### 3.1.1. Training Data

Ultrasound images were downsampled to fit in an image of 320 × 240 pixels. Where the original image was not 4:3 aspect ratio, it was letterboxed to lie centrally, and a black background added. The images were encoded using H.264 (greyscale, rate factor 23, zero latency, and YUV_4_2_0 palate) to provide a compact data storage with minimal loss. The original images had vertical heights of 80, 90, or 100 mm, and after letterboxing, the output images had vertical heights of 100–125 mm, leading to a pixel resolution of approximately 0.4–0.5 smm per pixel. It is worth noting that the axial resolution in mm of a 3 MHz 3-cycle ultrasonic pulse is 3 × 0.5 × 1540/3,000,000 × 1000 = 0.77 mm so our 320 × 240 image has better resolution than the underlying data. Preliminary tests indicated that, compared to a 320 × 240 video, a 800 × 600 video took 5× longer to analyse and a 1200 × 900 video took 12× longer. Therefore, 320 × 240 was determined to be a practical resolution.

The tongue contour may be partly obscured by mandible or hyoid shadows or otherwise indistinct. The labeller then has a choice either to omit keypoints in these regions or to estimate their position based on clues elsewhere in the image or audio. For this paper, we mainly adopted the latter approach.

Hand-labelling was carried out on 20 frames each, from 26 recordings. The frames were selected by k-means clustering using the DLC labelling tool so that they were distinct from each other. A few frames were rejected if they had no discernible features leaving a total of 520 test frames. The recordings comprised:A total of 10 recordings from 6 TaL Corpus [32] adult speakers (Micro system, 90° FOV, 64-element 3 MHz, 20 mm radius convex depth 80, 81 fps). These recordings were the first few recordings from the corpus and not specially selected.A total of 4 recordings from 4 UltraSuite corpus [33] Ultrax typically developing children (Ultrasonix RP system, 135° FOV, 128 element 5 MHz 10 mm radius microconvex, depth 80 mm, 121 fps). These were randomly selected. 10 recordings of the authors, using the Micro system with 64-element, 20 mm radius convex probe, and with different field of view and contrast settingsA total of 2 recordings by Strycharczuk et al. [34] using an EchoB system with a 128-element, 20 mm radius convex probe. These data are from an ultrasound machine not represented in the test set and included to generalize the model.

#### 3.1.2. Test Data

Hand-labelling was carried out on 40 k-means selected frames from 25 recordings using the DLC labelling tool to generate a total of 1000 test frames. Each recording was from a different speaker and were taken from several publicly available corpora:A total of 10 TaL corpus adult speakers (Articulate Instruments Micro system, 90 FOV, 64-element 3 MHz, 20 mm radius convex depth 80, 81 fps).A total of 6 UltraSuite Ultrax typically developing children (Ultrasonix RP system, 135° FOV, 128 element 5 MHz 10 mm radius microconvex, depth 80 mm, 121 fps).A total of 2 UltraSuite Ultrax speech sound disordered children (recorded as previous).A total of 2 UltraSuite UltraPhonix children with speech sound disorders (SSD) (recorded as previous).A total of 2 UltraSuite children with cleft palate repair. Ultrasound (Articulate instruments Micro system, 133° FOV, 64-element 5 MHz, 10 mm radius microconvex, depth 90, 91 fps.A total of 3 UltraspeechDataset2017 [35] adults. Ultrasound images (Terason t3000 system, 140° FOV, 128-element, 3–5 MHz 15 mm radius microconvex, depth 70 mm, 60 fps).

None of the test speakers were used in the training set. The hand-labelling was conducted by the first author with the same protocol used to train the DLC model (see Section 3.1.3). The second author also hand-labelled 25% of the same test frames (every fourth frame) for the purpose of comparing hand-labelling similarity.

#### 3.1.3. Ultrasound Keypoint Labelling

Eleven points were selected along the upper surface of the tongue: vallecula, root1, root2, back1, back2, dorsum1, dorsum2, blade1, blade2, tip1, tip2. This number is sufficient to describe the shape of the surface. Separation between the consecutive blade and tip points were approximately half that of other points in order to better represent the flexibility of that part of the tongue. An attempt was made to maintain consistency in placement relative to the tongue surface, even when these points were obscured by hyoid or mandible shadow. This approach to labelling differs from traditional labelling, which is limited in length to the extent of the bright edge visible in the image. In addition, keypoints were labelled on the hyoid and on the mandible at its base and at the mental spine where the short tendon attaches. The latter point is important as it forms the insertion point for the fanned genioglossus muscle fibres. These fibres principally control the midsagittal shape of the tongue body. Figure 2 shows the location of the labelled keypoints.

The bright surface of the epiglottis and any saliva bridge between the tip of the epiglottis and the tongue root is often traced as part of the tongue (see Figure 3A for an example). This may be an appropriate contour to trace if the boundary of the oral cavity is being assessed, but for studies investigating tongue root retraction and for the sake of consistently modelling the tongue surface, in this study, we elected to follow the surface of the tongue to the vallecula rather than the visible surface of the epiglottis.

It is sometimes the case that there are two apparent edges (e.g., Figure 3B). This most often occurs at the tongue blade when it is grooved to produce an (s) sound. In this study, we hand-labelled the lower of the two edges even when it was less distinct, as this represents the contour of the midline of the tongue.

Where possible, the tongue tip position was estimated even when there was no bright contour. In particular, the bright artefact often generated by the tip raising gesture was not labelled as part of the tongue contour. This bright artefact is due to the ultrasound beam reflecting off the surface of the tongue to the underside of the blade and back along the same path (Figure 4). Again, this means that the hand labels do not strictly follow the brightest edge.

Hyoid, mandible base, and short tendon base are indicated in Figure 4 as points A, B, and C, respectively.

### 3.2. EMA-Ultrasound Test Data

Simultaneously recorded ultrasound and EMA data are rare. Some pilot data generously made available by Manchester and Lancaster Universities (UKRI grant AH/S011900/1) were used here to evaluate the ability of DLC output to emulate EMA sensor movement. EMA data were recorded using the Carstens 501 system (Carstens Medizinelektronik GmbH, Bovenden, Germany) with three coils placed on the tongue tip (1 cm from apex), tongue blade and dorsum (approximately 15 mm separation between each sensor). The corpus consisted of three carrier phases “She said X clearly”, “She said X”, and “She said X again”. Unfortunately, in many of the ultrasound recordings there was loss of tip information as the probe failed to make contact with the chin. As a result of the restricted vocabulary and missing tip images, only five recordings were used to evaluate the ability of DeepLabCut to estimate EMA sensor movement. These were all the phrase “She said X clearly” with X = Bide, Bart, Bore, Bead, Bee.

### 3.3. Lip Camera Data Preparation

#### 3.3.1. Training Data

The TaL sample corpus was used for training. A total of 24 recordings, one from each of 24 speakers, were selected at random. Moreover, 7 or 8 frames were selected by k-means clustering from each recording, providing 207 training frames.

#### 3.3.2. Test Data

Testing was carried out on 10 TaL corpus speakers who were not included in the training set. A total of 40 frames were labelled from each speaker, providing 400 test frames. These speakers were selected to represent a range of age, sex, ethnicity, and facial hair. A second labeller re-labelled every fifth hand-labelled frame (20% of the test set).

#### 3.3.3. Lip Keypoint Labelling

The lip video was taken from the TaL corpus [32], which uses a front facing camera. This presents the opportunity to label the commissures (corners), midsagittal point on the upper and lower mucosal boundary, and points midway between the commissures and midpoints, as indicated in Figure 5.

### 3.4. Accuracy Measures

To assess the accuracy of an estimated tongue contour, Jaumard-Hakoun et al. [15] proposed a mean sum of distances (MSD). For every point on the hand-labelled curve, the distance to the closest point on the estimated curve was calculated, and for every point on the estimated curve, the distance to the closest point on the hand-labelled curve was calculated. The total sum of these distances was then divided by the total number of distance measures. This per-frame mean sum of distances was then averaged across all frames to give a single score. This curve similarity measure can give an identical score for the case where two curves match perfectly but one is longer than the other, and the case where the curves are the same length but do not match. We chose to separate these two factors. We used the standard MSD calculation, but when considering the endpoints of spline B, only one distance to a point on Spline A was included in the calculation: the shortest one. This means that the MSD is not affected by the relative lengths of the splines. Instead, we report a separate spline length difference measure (length hand-labelled spline—length estimated spline). Each spline was cubically interpolated prior to performing this MSD calculation so that it had 100 points regardless of its length.

DeepLabCut includes a root mean square error (RMSE) for the distance between hand-labelled points and estimated points. This measure is only applicable to DeepLabCut, since it is the only point estimation algorithm being considered.

For lip analysis, the area bounded by the upper and lower lip contours was measured in mm^2^. The width between commissures was measured. The MSD of the upper and lower lips were reported separately.

For comparison between EMA tongue sensors and the estimated tongue tip, blade, and dorsum keypoints, a Pearson correlation coefficient was calculated.

### 3.5. Ultrasound Tongue Contour Estimation Methods

The configuration of DeepLabCut, *SLURP*, *MTracker,* and *DeepEdge* are described in the following sections and Table 2 summarises their training and analysis rates.

#### 3.5.1. DLC Ultrasound

For body tongue/hyoid/mandible inference, we used DeepLabCut (version 2.1.10.0) [22,36]. We used a MobileNetV2-1.0 [25] based neural network with default parameters *. We also compared this with ResNet50, ResNet101, and EfficientNetB6 [25]. A total of 0.8 million iterations were used for training after preliminary testing (see Appendix C) showed convergence occurred with this amount of training. We validated with one held-out folder of the 1000 hand-labelled test frames. The image size was 320 × 240; ~0.5 mm/pixel. We then used a p-cut-off of 0.6 to determine root mean square error scores. This network was then used to analyse each of the test videos generating csv files of keypoints with associated confidence values, which were imported into the AAA software package (version 219_06, 2021, Articulate Instruments Ltd., Musselburgh, UK). The 11 tongue keypoints were converted into a single cubic tongue spline with 11 control points. The pixel to mm scale was calculated separately for each recording. MSD and length measures in millimetres were then made with respect to the hand-labelled keypoints similarly imported.

* ImgAug with ±25° rotation and random scaling in the range 0.5–1.25 (40% of the original dataset); pos_dist_threshold of 17.

#### 3.5.2. SLURP

The GetContours GUI [37] implemented in MATLAB was used to run the *SLURP* edge detection function. *SLURP* employs tongue-shape models but does not provide tools for in-domain training. Retraining the shape models on the training data used in this study was thought unlikely to make a substantial difference to the performance. Two different shape models provided by the author were tested and the one that gave the best results was selected. Increasing the minimum number of particles did not substantially improve the performance. The resulting reduction in the analysis rate was therefore not justified and the default settings were used:Colormap = “gray”, Sigma = 5.0, Delta = 2.0, Band Penalty = 2.0, Alpha = 0.80,Lambda = 0.95, Adaptive Sampling = Enabled, Particles = Min 10, Max 1000.

Each frame was seeded by hand with a 15-point spline. The tracker ran at 8.5 fps producing 100 edge contour points. These spline points were downsampled to 50 by removing every alternate point and imported into AAA software for MSD and length analysis.

#### 3.5.3. MTracker

A region of interest was defined as the area between the coordinates [50,50] and [200,300] relative to the top-left of the image. Dense U-Net model “dense_aug.hdf5” was used. This model has 50% of the training data with image augmentation. The tracker ran at 27 fps producing 100 points or fewer when the confidence threshold of 50% was not reached. These spline points were downsampled to 50 by removing every alternate point when imported into the AAA software for MSD and length analysis.

#### 3.5.4. DeepEdge

*DeepEdge* version 1.5 ran under MATLAB R2021a with deep learning toolbox, image processing toolbox, and computer vision toolbox. Three optional models are provided, each trained on different datasets. A model trained on the same ultrasound system and probe used in our 6 Ultrax TD child test recordings was tried first. However, this model performed more poorly on the test set than another of the models. The best performing model was (“DpEdg_CGM-OPUS5100_CLA651_21JUL2021”), and this was the model used for this study. All videos were mirrored, such that the tongue was pointing to the left, then after running *DeepEdge* and exporting the data, the results were then mirrored again to face tongue tip right before importing into AAA. The tracker ran at 3 fps producing 20 edge contour points. These contour points were imported into AAA software for MSD and length analysis.

### 3.6. Method for Comparing EMA Position Sensors to DLC Keypoints

The ultrasound data were analysed using the DLC ResNet50 model trained as per Section 3.5.1. The estimated tip1, blade1, and dorsum1 keypoints were picked as close matches to the tip, blade, and dorsum EMA sensors. Sections of the five recordings corresponding to the spoken utterances were selected. The sensor positions were compared to the estimated keypoints for every ultrasound frame timepoint, and Pearson correlation values recorded.

### 3.7. Method for Evaluating DLC Performance on Lip Camera Data

The same DeepLabCut configuration used for ultrasound images was used to train lip images. Only MobileNetV2_1.0 and ResNet50 encoders were tested. Keypoints were imported into the AAA analysis package (version 219_06, 2021, Articulate Instruments Ltd., Musselburgh, UK) as an upper lip spline and lower lip spline. Both splines shared the commissure keypoints as endpoints. MSD values were calculated for the upper lip and lower lip separately. A width (distance between commissure keypoints) and aperture (area enclosed by the upper and lower lip splines) were also recorded, as these are measures that speech scientists are interested in.

## 4. Results

### 4.1. Ultrasound Contour Tracking

We evaluated DLC with the MobileNetV2_1.0 encoder by training on 100% (twice), 75%, 50%, and 25% of the 520 hand-labelled frames. A small difference in MSD scores occurred between models generated in two separate training runs with 100% of the training data. This is likely due to the random selection of frames for image augmentation and the random amounts of augmented scaling and rotation. Table 3 shows that, compared to using 100% of the data, using 75% (390 frames) did not reduce performance significantly. ResNet50 backbone also produced no significant difference when using 75% compared to 100% of the training data. Using 50% of the total available hand-labelled frames, i.e., 260 hand-labelled frames with distinct tongue shapes extracted from 26 recordings, gave marginally poorer performance (*p* = 0.03). Performance was reduced when the number of frames used to transfer learning from human pose estimation to ultrasound tongue images was limited to 130. For this paper, we used models trained on all 520 hand-labelled frames. While it is possible that generalization to a very different scanner and probe might require the model to be retrained with supplementary frames from that domain, very few additional images would be needed. Certainly, no more than 260 and likely less than 50. The test set used here included recordings from a Terason scanner and probe unseen in the training set and performed very well (mean MSD 1.00 SD 0.30 for speaker TH c.f. mean MSD 1.06 SD 0.71 for all 25 speakers.

MSD mean and standard deviation values reported in Table 4 show *SLURP*, *MTracker,* and *DeepEdge* all performed less well on the test set used for this study than previously reported (1.7, 1.1 c.f. 2.3, 1.5) (1.4, 0.7 c.f. 3.2, 5.8) (1.4, 1.4 c.f. 2.7, 3.1). DeepLabCut still performed better than the originally reported MSD values for these other methods. 0.9 mm vs. 1.4–1.7 mm.

The quality of the ultrasound images may have been poorer in this test set than the original *SLURP*, *MTracker,* and *DeepEdge* studies, partly explaining the reduction in performance. It may also be the case that the training and test sets in the original studies were closely matched and the trackers have a limited ability to generalise to unseen data. In particular, *DeepEdge* comes with three models, each trained on a different system rather than one general model. If *DeepEdge* were trained on the same dataset that DeepLabCut was trained on it may have performed better, but DeepEdge requires at least 4× the available hand-labelled frames to train successfully and no training software is provided. Furthermore, the hand-labelled contours, used here as ground truth, follow the tongue contour and not necessarily the brightest edge. The original studies may have been evaluated against hand-labels of the brightest edge.

Table 4 shows that DLC with a ResNet50 encoder provided MSD scores equivalent to the MSD between the two hand-labellers in this study. The inter-labeller mean of 0.96 mm is close to the inter-labeller MSD of 0.9 mm reported by Jaumard-Hakoun [15]. It also indicates that, while the second hand-labeller tended to assign tongue contours 4% shorter than the first labeller, DLC was closer in length producing contours that were on average only 2% longer. The slightly poorer performance of ResNet101 compared with ResNet50 may be due to overtraining or variance in performance vs. number of training iterations (see Appendix C).

Table 5 shows the mean root mean square error scores across all keypoints with a confidence greater than 0.6 (60%). For example, if a tongue tip keypoint is obscured by the mandible shadow, then the network might generate a low confidence in its position and this point would be ignored. Using MobileNetV2 with 520 training samples as a baseline, the RMSE pixel accuracy is shown to decrease by up to 3.5% when less training data are used and increase by up to 3% when using a ResNet encoder. Interestingly ResNets are less accurate when all keypoints are considered but more accurate when unconfident points are ignored. EfficientNetB6 performed poorly, perhaps because the amount of training data were insufficient for such a large encoder network.

Figure 6 shows two frames evaluated within DLC. Image (a) shows that, although the 11 tongue keypoints hug the tongue surface, producing a low MSD value, they sometimes do not match the hand-labelled locations along that surface. This leads to RMSE scores of 6 pixels (~3 mm) compared to only 1 mm for MSD. Figure 7 and Figure 8 show how the overall results in Table 4 break down across test speakers. Speakers 01F_BL1 and PB both have very poor image quality, with DLC ignoring keypoints in some frames, resulting in shorter length estimates.

Figure 9 shows plots of *x*-axis = MSD vs. *y*-axis = %length difference for every test frame that generate the overall results in Table 4. An ideal estimator would have all points at (0,0) (see Appendix B for why an MSD of 0.0 is unlikely). Of the three previously reported estimators, *SLURP* is the most robust when MSD and length are considered. The tighter cluster for DLC more closely matches the inter-labeller plot.

Figure 10, Figure 11 and Figure 12 show every fifth hand-labelled test frame for the speakers 17ms (from the TaL corpus), 03F_BL1 (from the UltraSuite UXSSD corpus) and DF (from the UltraSpeech corpus). *SLURP* (green) has pretrained shape models, which restrict the shape of the contour. In Figure 10, a plausible tongue shape does not always match the underlying data. The flick upwards at the root of the tongue in Figure 11 may be as a result of how *SLURP*’s shape model was trained. *MTracker* (yellow) fits the tongue surface quite well, but because the length is controlled by a 50% confidence threshold, it very often omits the more difficult tip and root sections of the tongue contour. When we raised the threshold, *MTracker* performed very poorly in these regions. *DeepEdge* (pink) tended to underestimate the length. The option to postprocess by applying *EdgeTrak* to the neural net output produced poorer results, and so is not reported here. DLC ResNet50 (cyan) matches the hand-labelled contour (blue) so well that, in many frames, it sits directly on top. Disagreements mainly occur at the root where the hand-labelled contour is often speculative.

### 4.2. Ultrasound-EMA Point Tracking

The splines were scaled in mm and consisted of the 11 tongue-surface keypoints. The TT1, TBl1, and TD1 keypoints were selected as being close to the positions of the three EMA sensors on the tip blade and dorsum, respectively. The bite plane [38] was recorded in both the EMA and ultrasound data (see Figure 13) and both sets of data were rotated so that the bite plane formed the *x*-axis.

Figure 14 shows the comparison of x and y EMA sensor positions (red) with the positions estimated by DLC ResNet50 (black) as the phrase “She said bead clearly” is spoken. It is apparent that there is very little correlation in the *x*-axis, while there is a modest correlation in the *y*-axis.

Figure 15 plots, each EMA coordinate vs. the corresponding DLC estimated the coordinate for every ultrasonic frame of the five simultaneous EMA/ultrasound recordings. Again, correlation is only good for the y-coordinates of the tip and blade. Table 6 shows that the Pearson correlation coefficients calculated across all ultrasound frames for the five recordings confirm the visual findings.

### 4.3. Camera Lip Tracking

Table 7 shows RMSE scores for all lip keypoints. Unexpectedly, ResNet50 does not significantly outperform MobileNetV2. It may be that although experiments on the ultrasound images showed that 260 frames were adequate for good MobileNetV2 performance, the 207 training frames used here were insufficient for ResNet50 to reach its full potential. We used fewer training frames because the training set was homogeneous as each speaker was recorded under identical conditions.

Table 8 shows a comparison of MSD for upper and lower lips, lip aperture and lip width for the two DLC encoders and the second labeller. As with RMSE, these performance indicators reveal that unlike for ultrasound images, DLC does not quite match the inter-labeller lip performance. More training data might improve the performance. Lip contouring does not follow a bright edge. Indeed, the lower lip contour labelling criterion was to mark where it would meet the upper lip rather than the visible boundary of the lip and oral cavity. As shown by the smaller average area estimates, DLC tends to mark this visible boundary of the lips and oral cavity. This is also apparent in the labelled images shown in Figure 16, Figure 17 and Figure 18. Estimation of the commissures and, therefore, of the width of the mouth is, however, as accurate as the inter-labeller score.

Figure 19 shows the MSD values for lower and upper lips, comparing DLC MobileNetV2, DLC ResNet50, and inter-labeller. Of note, ResNet50 improves the estimates of speaker 12me, but it performs more poorly on speaker 17ms (their lips are partially obscured by a moustache). Example frames from these two speakers are shown in Figure 17 and Figure 18, respectively. MobileNetV2 estimates the lower lip closer to the lip edge for speaker 12me than the labeller. For speaker 17ms, ResNet50 can be seen to perform very poorly and with low confidence.

Figure 20 and Figure 21 show the lip aperture and width respectively, comparing DLC MobileNetV2, DLC ResNet50 and inter-labeller for each speaker. As can be observed in Figure 19, ResNet50 had trouble identifying the lip commissures for speaker 17ms, but MobileNetV2 did surprisingly well. These figures also show that the second human labeller had trouble following the instructions for the positioning of the lower lip boundary for speakers 07me and 12me. They also overestimated the width in speaker 12me, where whiskers obscured the commissure positions. High Pearson correlation values of 0.95 for hand-labelled vs. DLC lip aperture and 0.89 for hand-labelled vs. DLC lip width were recorded for DLC MobileNetV2.

## 5. Discussion

In this study, we investigated an open tool for pose estimation applied to speech articulator image data. By leveraging existing networks pretrained on general object recognition and human body pose estimation, relatively small amounts of speech articulator training data result in a model capable of achieving human-level accuracy on unseen data. In a field dominated by segmentation and edge-detection methods, we show the following for the first time:Pose estimation is capable of learning how to label features that do not necessarily correspond to edges.Pose estimation can estimate feature positions to the same level of accuracy as a human labeller.

The hand-labels used here, as in similar studies, were subjectively determined. However, labelling, adhering to the prescribed guidance (see Section 3.1.3), could be learnt by both another human and the DLC network equally well. From this, we can be encouraged that if a more principled ground truth were to be established, perhaps by mapping points from MRI or EMA onto the ultrasound images, then that ground truth would be learnt.

It is possible that the performance of *SLURP*, *MTracker,* and *DeepEdge* could be improved if trained on the same data as DLC. However, Zhu et al. [11] test *MTracker* on two of the test sets used here; namely, the Ultrax Child corpus and the French UltraSpeech corpus. Looking at Figure 7 and Figure 8, speakers from those datasets (02TD1M, 05TD1M, 07TD1F, 08TD1F, 09TD1, 11TD1F, TH, DF, PB) do not show broadly better performance by *MTracker*. Conversely, the UltraSpeech corpus is not represented in the DLC training data and Figure 7 and Figure 8 do not show worse performance by DLC on TH, DF, and PB than on other speakers. Neither *DeepEdge* nor *MTracker* make a training package publicly available. If users could train these models on their data, it is questionable whether they would choose to hand-label the 2000–35,000 frames needed to train these networks. By contrast, the DLC model trained in this study appears to generalise well. DLC includes a simple training package and the training data used in this study is available online. Thus, if the model did perform poorly on a user’s dataset, a few (<100) hand-labelled frames from that dataset could be added to the existing training data and the model retrained. The small number of images required for training also permits time for more careful, consistent, and expert labelling.

Outside the scope of this study, DeepLabCut can also estimate the position of the hyoid, and jaw if these features lie within the image. These are point structures rather than edges and cannot easily be estimated by segmentation or edge detection algorithms.

Where speed is a consideration, DLC MobileNetV2 and ResNet50 perform faster (with a GPU) than real time even with ultrasound frame rates of 119 fps. DLC could therefore perform tongue contour estimation on live ultrasound and lip images. Real-time performance is important for live lipreading or visual feedback of tongue for speech therapy. For offline analysis, DLC MobileNetV2 performs at 7 fps using a CPU and can process a batch of recordings at this rate. It does not require manual intervention for each recording so can be left to run overnight if necessary.

DeepLabCut analyses each frame independently. No frame-to-frame continuity is applied. Given that it tracks so well, the absence of temporal continuity constraints can be seen as an advantage because problems of “drift” in contour position cannot occur. Frame-to-frame jitter in keypoint position can be filtered out in post-processing if the frame rate is significantly faster than the articulator movement.

Pose estimation offers the possibility of tracking keypoint positions. Whether it is possible to track points on the tongue remains an open question. Results from our pilot investigation comparing the EMA sensor position to DLC estimates, and high RMSE values (~3 mm) w.r.t. MSD values (~1 mm), both indicated poor estimation of the sensor position along the tongue surface. This is likely due, in part, to inconsistency of training keypoint placement by the human labeller, despite an effort in this study to try to label as if the keypoints were attached to a specific flesh point. A further multi-speaker study where simultaneous EMA and ultrasound is used to train and to evaluate the estimation of the sensor positions is required.

Edge tracked partial tongue contours provide no indication of which part of the contour corresponds to root, body, or tip. This has dictated what kind of further analysis can be performed on estimated lip and tongue contours. The intersection of the tongue contour with a fixed measurement axis is often used to assess raising or lowering of a part of the tongue. Measuring tongue tip movement in this way runs into problems when the tip is retracted, and the contour no longer crosses the axis. Pose estimation opens the possibility of measuring the contraction of the root, body, dorsum, and blade with respect to the anatomically defined position of the short tendon where bundles of the genioglossus muscle attach to the mandible (see Figure 2). This provides a measure independent of probe rotation. Lip rounding can be identified not only using the overall width and aperture measures but also the relative height of the midpoints compared to the parasagittal points. With reference to Figure 5, a measure for lip rounding can be formulated as:(abs (C − D) − 0.5 (abs (E − F) + abs (G − H))) / abs (A − B)

Pose estimation has recently been applied to sustained speech articulations recorded using MRI of the vocal tract [39]. A total of 256 × 256-pixel images with 1 pixel/mm resolution were analysed and RMSE accuracy results of 3.6 mm reported. These results are similar to the RMSE scores reported here for ultrasound. Beyond the scope of the current study, we piloted articulatory keypoint estimation using dynamic MRI of the vocal tract taken from a public multi-speaker dataset [40]. The data consisted of 84 × 84-pixel images (83 Hz) and perhaps because of the low spatial resolution, the method was less successful. A larger amount of training data were perhaps required, and this would be something to be investigated further.

DeepLabCut provides a package for estimating 3D positions using multicamera data and could be applied to form a richer feature set for lip movement. A side-facing camera would capture lip protrusion information. DeepLabCut could also be investigated as a means for tracking other expressive facial features, such as eyebrows, or for monitoring head movement. DeepLabCut also provides a package to run on a live video stream and work is underway to implement this for live ultrasound input.

In summary, the combination of transfer learning and pose estimation, evaluated here using DeepLabCut, provides a ground-breaking level of efficiency, practicality, and accuracy when applied to feature labelling of speech articulatory image data. The models generated by this study have been made available in Appendix A for use and further evaluation by other research groups.

## Figures and Tables

**Figure 1 sensors-22-01133-f001:**
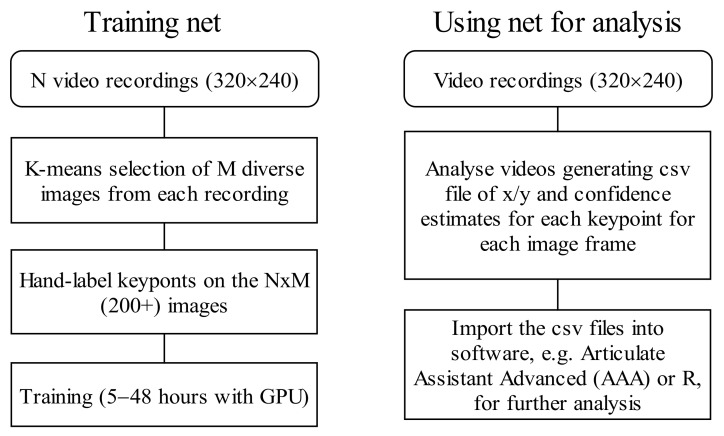
DeepLabCut training and analysis processes.

**Figure 2 sensors-22-01133-f002:**
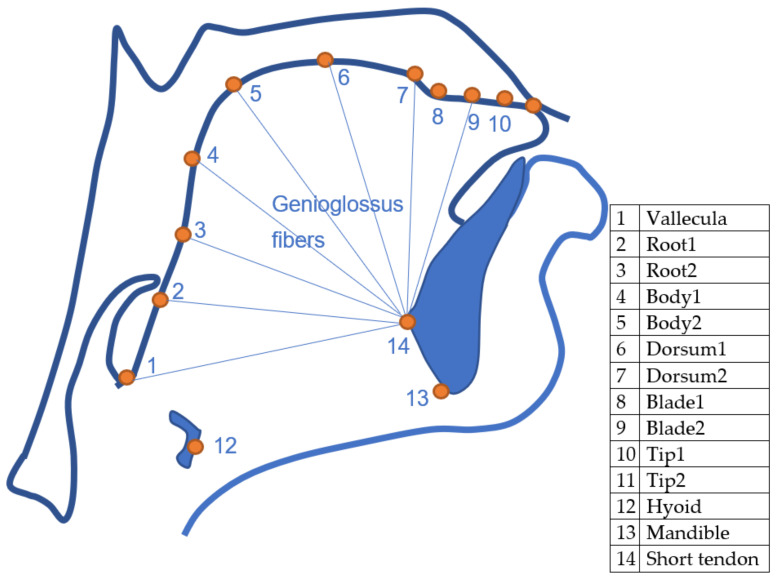
Outline of midsagittal tongue contour and the labelled keypoints.

**Figure 3 sensors-22-01133-f003:**
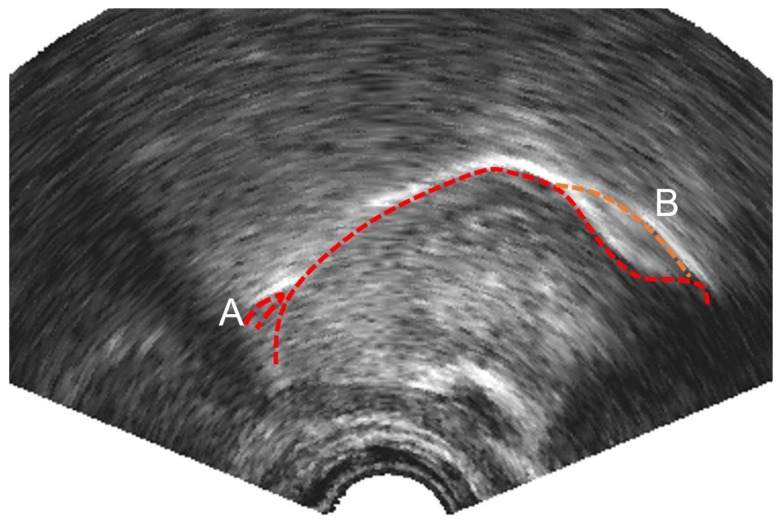
Ultrasound image showing (**A**) bright reflection from tip of epiglottis (**B**) double reflection parasagittal surface (upper) and midsagittal surface (lower) of the tongue blade.

**Figure 4 sensors-22-01133-f004:**
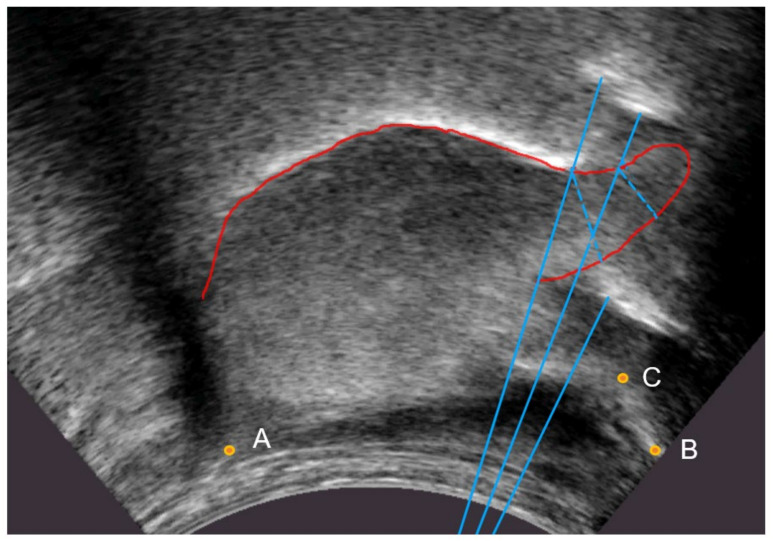
Ultrasound image with beam tracing (blue) showing actual path of ultrasound beam (dotted) and the resulting bright artefacts based on the equivalent time of travel in the direction of the transmitted beam (solid). A—hyoid; B—mandible base; C—short tendon base.

**Figure 5 sensors-22-01133-f005:**
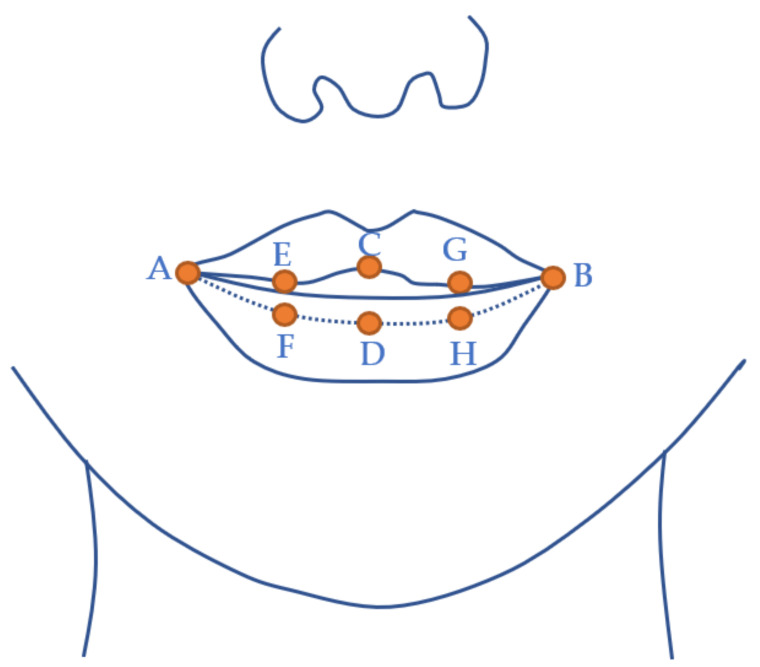
Keypoints labelled on the lip video. Dotted line indicates mucosal border on the lower lip that makes contact with the upper lip when the mouth is closed. A and B—commissures; C and D—centre of the upper and lower lip, respectively (defined by philtrum midpoint and not necessarily equidistant from A and B); E—equidistant from A and C; F—equidistant from A and D; G—equidistant from B and C; H—equidistant from B and D.

**Figure 6 sensors-22-01133-f006:**
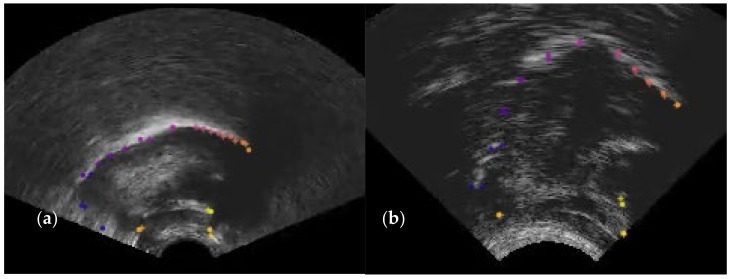
(**a**) Shows points estimated to lie on the tongue surface but distributed differently to the hand labels; (**b**) an example where the positions are estimated accurately. ‘+’ indicates the estimated position.

**Figure 7 sensors-22-01133-f007:**
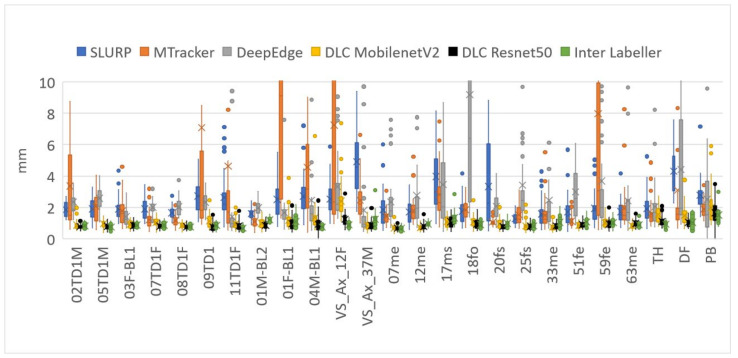
Mean sum of distances (MSDs) between hand-labelled randomly selected frames (40 frames for each speaker) and each of the assessed methods (DLC using MobileNetV2_1.0 encoder).

**Figure 8 sensors-22-01133-f008:**
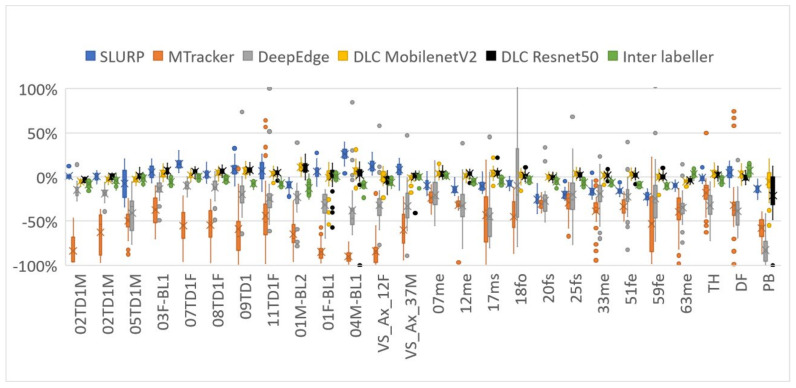
Relative length distance between hand contoured randomly selected frames (40 frames for each participant) and each of the assessed methods (DLC using MobileNetV2_1.0 encoder).

**Figure 9 sensors-22-01133-f009:**
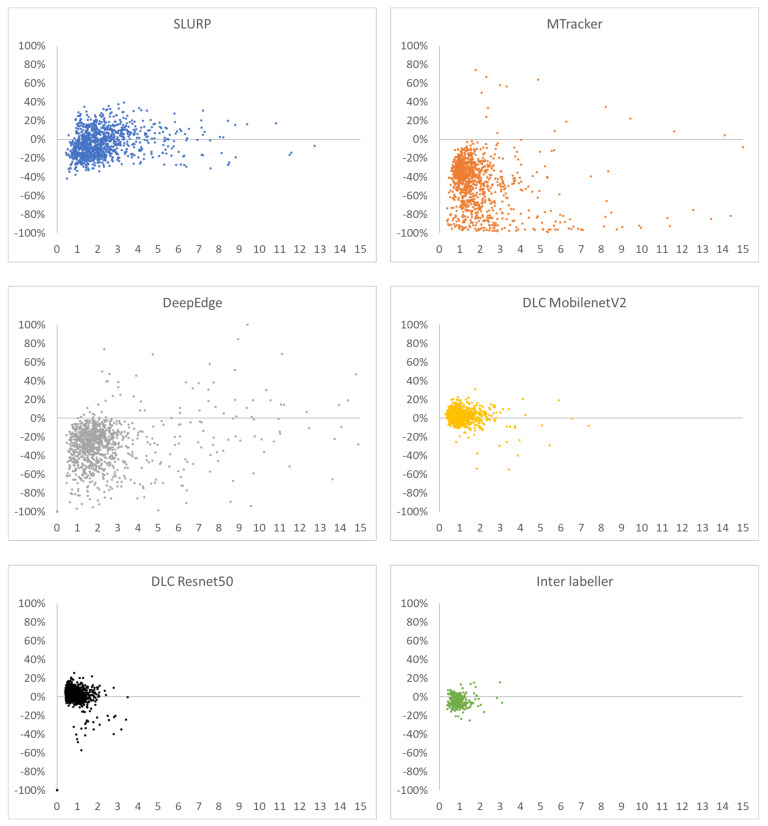
MSD vs. %length difference (hand-label estimator) for SLURP, MTracker, DeepEdge, DLC MobileNetV2, DLC ResNet50, and second labeller.

**Figure 10 sensors-22-01133-f010:**
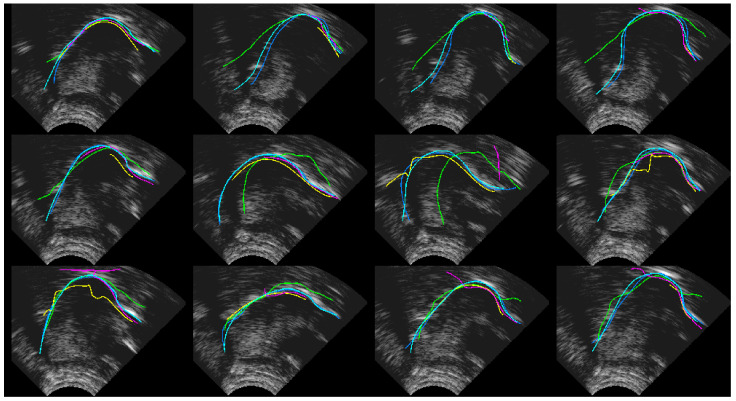
Speaker TaL17ms; blue—hand-label; green—SLURP; yellow—MTracker; pink—DeepEdge; cyan—DLC_ResNet50.

**Figure 11 sensors-22-01133-f011:**
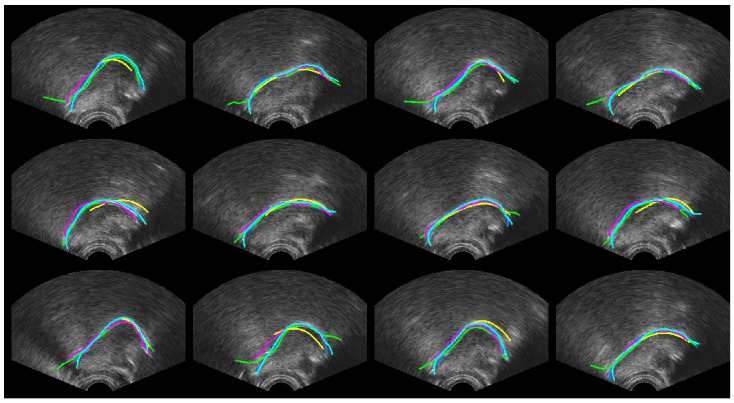
Speaker UXSSD03F; blue—hand-label; green—SLURP; yellow—MTracker; pink—DeepEdge; cyan—DLC_ResNet50.

**Figure 12 sensors-22-01133-f012:**
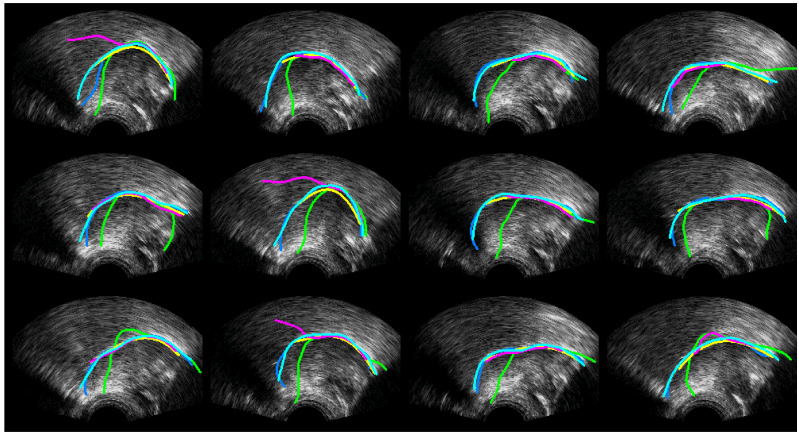
Speaker UltraSpeechDF; blue—hand-label; green—SLURP; yellow—MTracker; pink—DeepEdge; Cyan—DLC_ResNet50.

**Figure 13 sensors-22-01133-f013:**
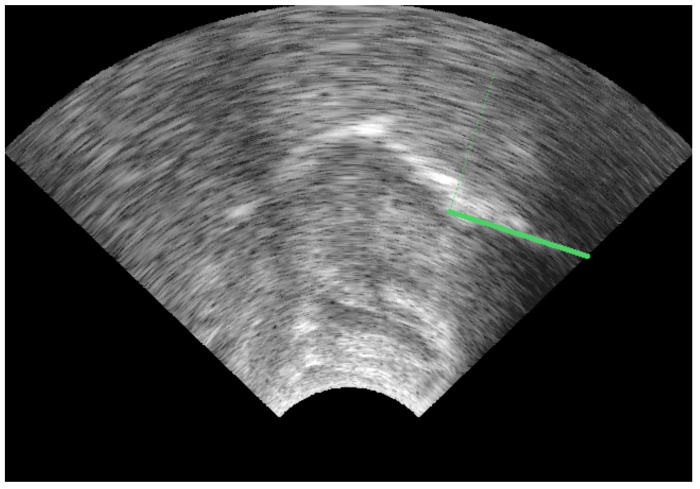
Image of the tongue pressed against a bite-plate and a green fiducial line superimposed. All coordinates were rotated so that the green line formed the horizontal axis.

**Figure 14 sensors-22-01133-f014:**
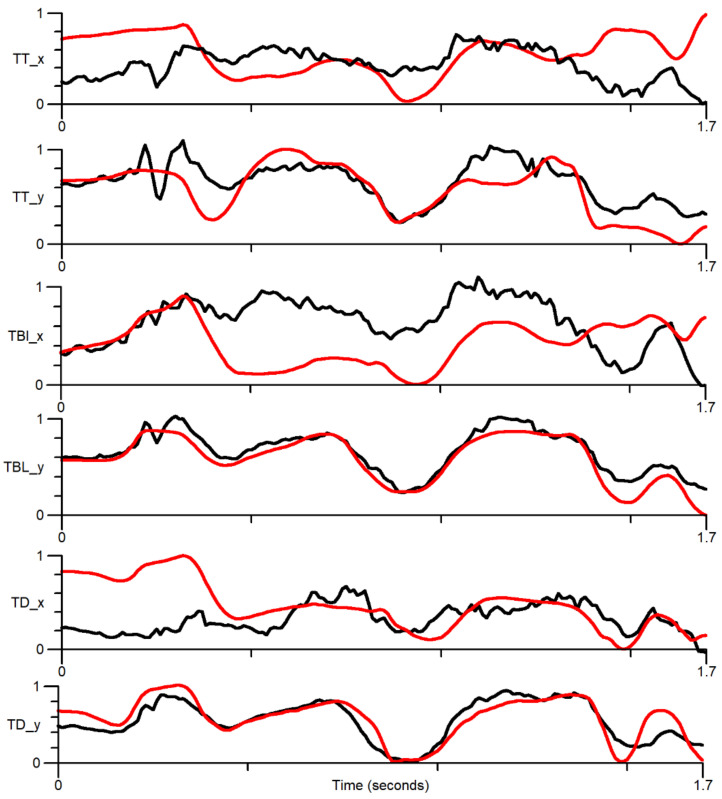
The phrase ‘She said “bead” clearly’ showing x and y position against time for the tongue tip (TT), blade (TBl), and dorsum TD. Red—EMA sensor; black—DLC estimated position. The *y*-axis has no units.

**Figure 15 sensors-22-01133-f015:**
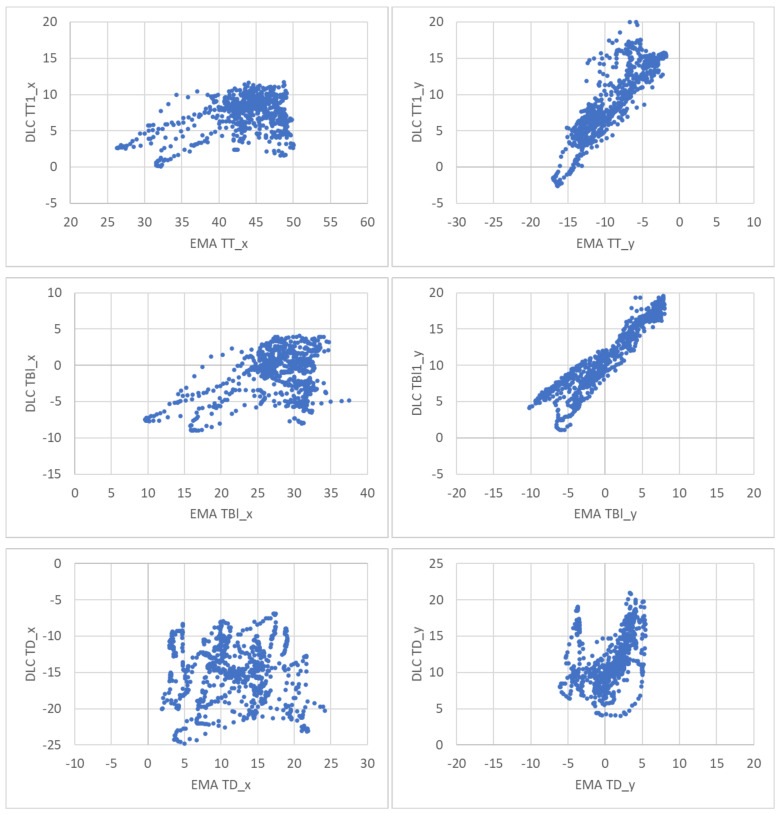
EMA coordinate vs. DLC estimated coordinate plotted for every ultrasonic frame of the 5 simultaneous EMA/ultrasound recordings. TT—tongue tip; TBl—tongue blade; TD—tongue dorsum.

**Figure 16 sensors-22-01133-f016:**
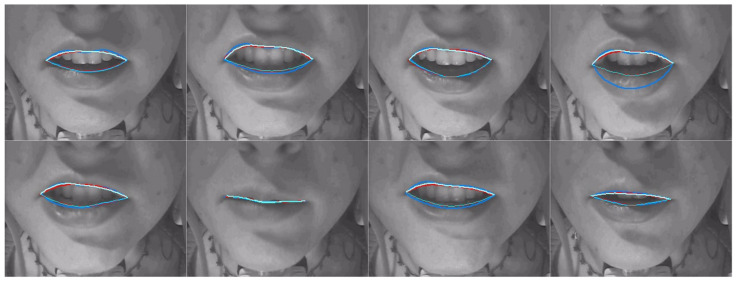
Speaker 25fs; blue—hand-label; red—MobileNetV2; cyan—ResNet50.

**Figure 17 sensors-22-01133-f017:**
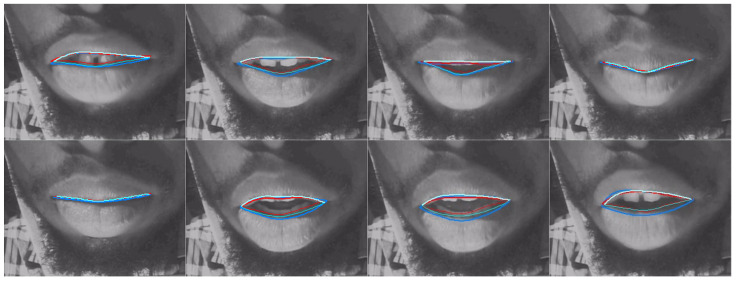
Speaker 12me; blue—hand-label; red—MobileNetV2; cyan—ResNet50.

**Figure 18 sensors-22-01133-f018:**
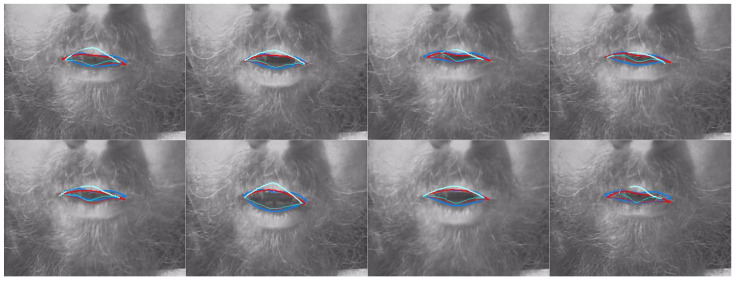
Speaker 17ms; blue—hand-label; red—MobileNetV2; cyan—ResNet50.

**Figure 19 sensors-22-01133-f019:**
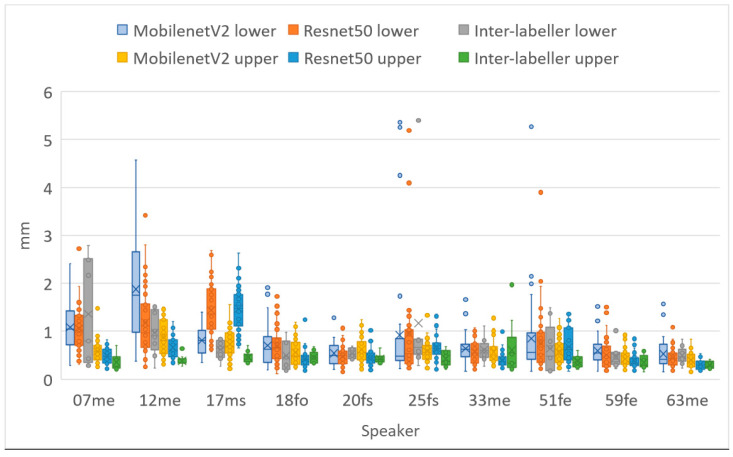
Mean sum of distances (MSDs) between hand-labelled randomly selected frames (40 frames for each speaker) and DLC using MobileNetV2_1.0 and ResNet50 encoders. Upper lip and lower lip shown separately.

**Figure 20 sensors-22-01133-f020:**
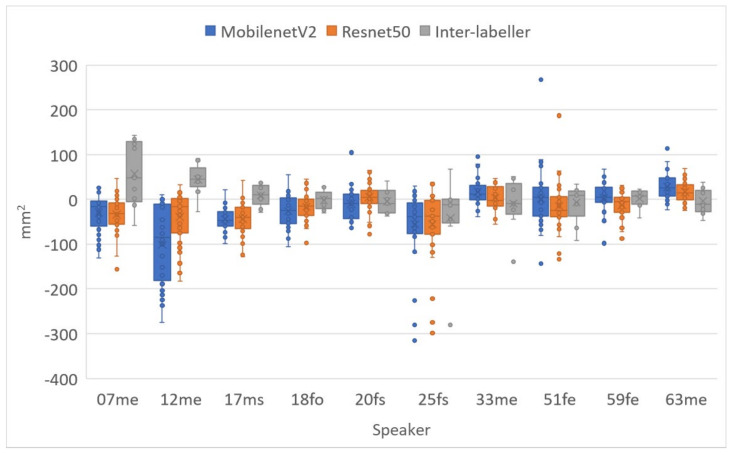
Difference in lip area between hand-labelled randomly selected frames (40 frames for each speaker) and DLC MobileNetV2_1.0, DLC ResNet50 and a second hand-labeller.

**Figure 21 sensors-22-01133-f021:**
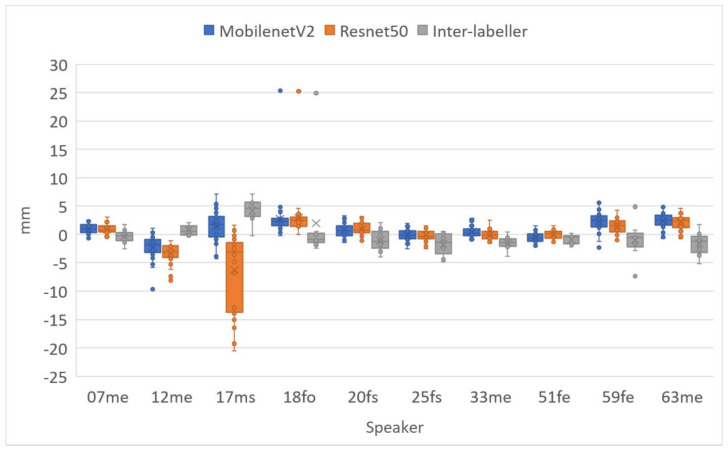
Difference in lip width between hand-labelled randomly selected frames (40 frames for each speaker) and DLC MobileNetV2_1.0, DLC ResNet50, and a second hand-labeller.

**Table 1 sensors-22-01133-t001:** Mean sum of distances (MSD) error scores reported in the literature for estimated vs. hand traced contours quoted by authors for speaker independent tests.

Algorithm	MSD (Mean/s.d.) mm
*EdgeTrak* ^1^	6.8/3.9
*SLURP* ^1^	1.7/1.1
*TongueTrack* ^1^	3.5/1.5
*AutoTrace* ^1,2^	2.6/1.2
*DeepEdge* (NN + Snake) ^3^	1.4/1.4
*MTracker* ^4^	1.4/0.7
*BowNet* ^5^	3.9/-
*TongueNet* ^5^	3.1/-
*IrisNet* ^5^	2.7/-
Human-human	0.9 ^6^/-, 1.3 ^7^

^1^ MSD values taken from Laporte et al. [5]. ^2^ when trained on the first 1000 frames of the test set. ^3^ MSD values taken from Chen et al. [12] 5.7 pixels at 0.25 mm/pixel. ^4^ MSD values taken from Zhu et al. [11]. ^5^ MSD values taken from Mozzafari et al. [9]. Value in mm estimated based on 128 × 128 images of 80 mm depth = 0.638 mm/pixel. Trained and tested on the same dataset with 5% test holdout. ^6^ Reported MSD between two hand-labellers Jaumard-Hakoun et al. [15]. ^7^ Reported RMSE standard deviation of 7 labellers Csapo and Lulich [4].

**Table 2 sensors-22-01133-t002:** Comparison of ultrasound contour tracking algorithms showing the analysis frame rate, image size, training frame rate, and time for the network training to converge.

Algorithm	Frames per Second ^1^ (GPU/CPU)	Image Size	Training Data/Time (Frames/Hours)
SLURP ^2,3^	NA/8.5	data	N/A
DeepEdge (NN + Snake)	2.7/NA	64 × 64	2700/2
DeepEdge (NN only)	3.0/NA	64 × 64	2700/2
MTracker	27/NA	128 × 128	35,160/2
DeepLabCut (MobNetV2_1.0)	287/7.3 ^4^	320 × 240	520/7.5
DeepLabCut (ResNet50)	157/4.0 ^4^	320 × 240	520/16
DeepLabCut (ResNet101)	105/2.6 ^4^	320 × 240	520/30
DeepLabCut (EfficientNet B6)	27/1.7 ^4^	320 × 240	520/48

^1^ Using Windows laptop PC with Core i7-10750H 16GB RAM and NVIDIA RTX 2060 MaxQ. ^2^ SLURP is the only algorithm tested here that does not use the NVIDIA GPU. ^3^ SLURP requires the first frame of each recording to be manually seeded with at least 6 points using GetContours MATLAB GUI. The timing recorded here excludes this manual labelling step. ^4^ Analysing using batch size 8.

**Table 3 sensors-22-01133-t003:** Error scores vs. hand contoured for 20, 15, 10, and 5 frames per recording used for training.

MobileNetV2 Training Data	MSD (Mean, s.d., Median)	MSD *p* Value ^1^	%Length Diff (Mean, s.d., Median)
conf 80% 520 frames	1.06, 0.59, 0.90	1.00	+1.8, 7.0, +2.0
	1.06, 0.71, 0.89	0.89	+1.8, 9.1, +1.8
conf 80% 390 frames	1.12, 0.86, 0.91	0.09	+2.8, 10.7, +2.5
conf 80% 260 frames	1.13, 0.71, 0.94	0.03	+1.9, 9.7, +1.7
conf 80% 130 frames	1.17, 0.79, 0.94	<0.001	+3.5, 8.1, +3.2

^1^ Two tailed *t*-test assuming equal variance with reference to the MSD data generated by the model corresponding to the first row MSD distribution.

**Table 4 sensors-22-01133-t004:** Error scores vs. hand contoured (including regions where hand labels had to be guessed at tip and vallecula.

Algorithm	MSD (Mean, s.d., Median)	MSD *p* Values ^1^	%Length Diff (Mean, s.d., Median)
SLURP	2.3, 1.5, 1.9	<0.001	−3.8, 14.4, −4.6
DeepEdge (NN only)	2.8, 3.1, 1.9	<0.001	−27.5, 25.3, −26.0
MTracker	3.2, 5.8, 1.5	<0.001	−49.0, 28.7, −44.4
DLC (MobileNetV2_1.0 conf 80%)	1.06, 0.59, 0.90	0.04	+1.8, 7.0, +2.0
DLC (ResNet50 conf 80%)	0.93, 0.46, 0.82	0.29	+1.6, 8.8, +2.2
DLC (ResNet101 conf 80%)	0.96, 0.67, 0.81	0.80	+1.8, 9.1, +1.8
Inter-labeller	0.96, 0.39, 0.88	1.0	−4.3, 6.2, −4.8

^1^ Two tailed *t*-test assuming unequal variance with reference to the MSD data generated by the inter-labelling.

**Table 5 sensors-22-01133-t005:** Root mean square error scores on test set and times for training.

Network	RMSE Test (*p* > 0.6) Pixels	Train Time ^1^ 0.8 Million Iterations	Analyse Time ^1^ Frames/s
DLC (MobileNetV2_1.0)	6.15	7.5 h	190
DLC (MobileNetV2_1.0)	6.17	7.5 h	190
DLC (MobileNetV2_1.0) 75%	6.28	7.5 h	190
DLC (MobileNetV2_1.0) 50%	6.39	7.5 h	190
DLC (MobileNetV2_1.0) 25%	6.38	7.5 h	190
DLC (ResNet50)	6.07	16 h	100
DLC (ResNet101)	5.99	30 h	46
DLC (EfficientNet b6)	11.55	48 h	14

^1^ Time measured using a GTX 1060 GPU (slower than the GPU used for timings in Table 2).

**Table 6 sensors-22-01133-t006:** Pearson correlation values for each sensor coordinate calculated over the five recordings.

Sensor Coordinate	Pearson Correlation Coefficient
Tongue tip x	0.37
Tongue tip y	0.88
Tongue blade x	0.39
Tongue blade y	0.93
Tongue dorsum x	−0.03
Tongue dorsum y	0.44

**Table 7 sensors-22-01133-t007:** Root mean square error (average for all lip keypoints).

Network	RMSE Test (*p* > 0.6) Pixels
DLC (MobileNetV2_1.0)	3.79
DLC (ResNet50)	3.74

**Table 8 sensors-22-01133-t008:** MSD, aperture difference, and width difference comparing hand labels to DLC (MobileNetV2_1.0).

Lip Measure	Inter Labeller Mean/s.d./Median	DLC MobileNetV2_1.0 Mean/s.d./Median (*p* Value) ^1^	DLC ResNet50 Mean/s.d./Median (*p* Value)
MSD upper lip (mm)	0.41/0.23/0.36	0.59/0.29/0.54 (<0.001)	0.59/0.40/0.47 (=0.001)
MSD lower lip (mm)	0.73/0.71/0.55	0.86/0.75/0.64 (0.17)	0.82/0.67/0.64 (0.65)
Lip aperture (mm^2^)	4.6/54/6.2	−23/61/−10	−19/48/−11
Lip width (mm)	−0.1/3.6/-0.5	0.8/2.4/0.7	−0.2/3.7/0.4

^1^ Two tailed *t*-test assuming unequal variance with reference to the MSD data generated by the DLC inter-labeller distribution. Not applicable to aperture and width.

## Data Availability

All files and folders used for DeepLabCut training and testing are available at https://github.com/articulateinstruments/DeepLabCut-for-Speech-Production (accessed on 28 November 2021).

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
