# Peer review of "Beyond the Edge: Markerless Pose Estimation of Speech Articulators from Ultrasound and Camera Images Using DeepLabCut"

_sensors, 2022, doi:10.3390/s22031133_

Round 1

Reviewer 1 Report

The papers presents an experimental analysis where an open tool for pose estimation based on neural network is used to estimate the pose of speech articulators without using traditional edge detectors approach. The paper experiments with different data and different tasks (MRI, lip tracking from images and EMA sensors) comparing against state of the art approaches.

I am not very familiar with this research area, so I cannot judge the amount of novelty in this work. However, in my opinion this paper presents a very interesting and detailed experimental study that can result useful to other researchers in the area. Clearly, as the author state, this study is limited in some parts and leaves several further research directions open. So, I would recommend accepting this paper.

However, I have some comments that I would like the Authors to address. In particular the comparison with the edge detection method is a major issue.

  • In Table 1 it is not clear on what data those numbers are computed. Later in the paper I realize that they are coming from different datasets. I have another comment about this later. Anyway, I would propose reporting the datasets when the papers are mentioned in the text
  • In general I think that there is some confusion about the data. The paper presents numbers and figures before the datasets are introduced in Sec 4. I would suggest discussing briefly about the data in the introduction, without many details of course. So that readers have an idea of what the figures and numbers refer to.
  • Still related to the data, I would suggest including a table that summarizes the different datasets being used in all the tasks.
  • In Sec. 4.1.1 explains how (and why) the 10, 4, 10 and 2 recordings are selected from the 4 datasets
  •  Results in Table 4 are not actually comparable. This is in my opinion a very major error. In its current form the comparison is not only useless, it is wrong! To compare results all approaches should be used in the same conditions. If SLURP, DeepEdge and MTracker require in-domain training material they should be retrained. Otherwise, my suggestion is to remove those methods to avoid confusion. The inter-labeller performance already set an upper bound for the task. The proposed approach reaches that upper bound so that's enough in my opinion.
  • In table 3 the Authors investigate the amount of training material needed to adapt the pose estimation model. The table refers only to 1 of the models available (MobileNet). I would ask the authors to discuss about what they would expect from the other models. The same behaviour?
  • For Figures 10 to 12 add a legend (at least in one of them, although I understand that it is the same as before, so it is not really necessary.  

Line 327: I think that the recordings are 26

Line 474 "75% 390 frames": check please

Author Response

Thank you for taking the time to review this paper.

The papers presents an experimental analysis where an open tool for pose estimation based on neural network is used to estimate the pose of speech articulators without using traditional edge detectors approach. The paper experiments with different data and different tasks (MRI, lip tracking from images and EMA sensors) comparing against state of the art approaches.

I am not very familiar with this research area, so I cannot judge the amount of novelty in this work. However, in my opinion this paper presents a very interesting and detailed experimental study that can result useful to other researchers in the area. Clearly, as the author state, this study is limited in some parts and leaves several further research directions open. So, I would recommend accepting this paper.

However, I have some comments that I would like the Authors to address. In particular the comparison with the edge detection method is a major issue.

  • In Table 1 it is not clear on what data those numbers are computed. Later in the paper I realize that they are coming from different datasets. I have another comment about this later. Anyway, I would propose reporting the datasets when the papers are mentioned in the text

Table 1. summarises the state-of-the-art and allows us to rationalise which current methods to compare to DeepLabCut pose estimation. We have reworded the first and last lines of the section containing table 1 to make it clearer that this table is compiled in order to make a decision about which state-of-the-art tongue tracking methods to compare with pose estimation.

  • In general I think that there is some confusion about the data. The paper presents numbers and figures before the datasets are introduced in Sec 4. I would suggest discussing briefly about the data in the introduction, without many details of course. So that readers have an idea of what the figures and numbers refer to.

We have moved the keypoint labelling discussion to section 4 after each dataset is introduced.

  • Still related to the data, I would suggest including a table that summarizes the different datasets being used in all the tasks. We have opted to complete a bulleted lists rather than introduce another table.
  • In Sec. 4.1.1 explains how (and why) the 10, 4, 10 and 2 recordings are selected from the 4 datasets We have added more explanation for why the recordings were selected from the 4 datasets. We hesitate to add too much information as another reviewer has requested that we shorten the paper.  Only 2 recordings were included from the last dataset because all the sentences in that corpus were very similar.
  •  Results in Table 4 are not actually comparable. This is in my opinion a very major error. In its current form the comparison is not only useless, it is wrong! To compare results all approaches should be used in the same conditions. If SLURP, DeepEdge and MTracker require in-domain training material they should be retrained. Otherwise, my suggestion is to remove those methods to avoid confusion. The inter-labeller performance already set an upper bound for the task. The proposed approach reaches that upper bound so that's enough in my opinion.

SLURP does not require in-domain training. Our discussions with the author indicated that it would be unlikely to make a substantial difference if the tongue shape models were trained on different data. Even so we tried two different shape models provided by the author and chose the one that gave the best results. We had extensive communications with the author to optimise the algorithm parameters. We had extensive discussions with the author of DeepEdge.  We sent our training data to the author but they intimated that they needed 4x the amount of available training data to train their model. They provide three optional models, each trained on different datasets. We initially tried a model they had trained on the same ultrasound system and probe used in our 6 Ultrax TD child test recordings. However, this model performed more poorly than another of their models. We used the best performing model.  The paper published by Zhu on MTracker tests their system on two of the test sets that we tested on. Namely, the Ultrax Child corpus and the French Ultraspeech corpus.  We are therefore replicating the authors work on the same out-of-domain test data as the authors used. Looking at Figures 7&8, speakers from those datasets (02TD1M, 05TD1M, 07TD1F, 08TD1F, 09TD1, 11TD1F, TH, DF, PB) do not show broadly better performance. Conversely, DeepLabCut had not seen data from the UltraSpeech corpus during training and Figures 7&8 do not show worse performance on TH, DF and PB than on other speakers. Neither DeepEdge nor MTracker make a training package publicly available. Part of the reason for this is that they require so much hand labelled training data as to make it impractical for users to train on their dataset.  Nevertheless, we feel that it is important to give readers an idea of how DeepLabCut performs in comparison to other available methods that report good results. 

  • In table 3 the Authors investigate the amount of training material needed to adapt the pose estimation model. The table refers only to 1 of the models available (MobileNet). I would ask the authors to discuss about what they would expect from the other models. The same behaviour? We have tested the resnet50 backbone with 75% of the training data and found the RMSE scores on the test set to be equivalent to using 100% of the data. We have not tested Resnet101 with less training data. We have reworked the Appendix B figure on RMSE scores against number of training iterations. From that it can be seen that Resnet101 may be slightly overtrained with 0.8 million iterations. We could add more training data to reduce that effect or stop the training earlier.
  • For Figures 10 to 12 add a legend (at least in one of them, although I understand that it is the same as before, so it is not really necessary.  We do not want to make these images smaller by adding a legend at the side. A legend at the bottom of the image would simply add a new line above the figure caption. We are not sure that this helps the reader.

Line 327: I think that the recordings are 26 Yes, thanks. There was an error which we have corrected. We have listed the training recordings so that it is clear that they add up to 26

Line 474 "75% 390 frames": check please Brackets have been added

Reviewer 2 Report

The paper mainly investigates the effect of using the models designed for pose estimation to estimate the speech articulator key-points without markers. This exploration is meaningful. But there are some problems with the manuscript.

(1) The writing is not so good and it's a little hard to read the manuscript. The main problem is that there are too many unnecessary long sentences. For example, the "-ing" phrase in line 260 is almost a complete sentence, which would be better to use a clause instead of the lengthy "-ing" phrase.

The single sentence in line 261-263 includes a second clause in the first attributive clause, which is really not comfortable to read this long sentence.

The sentence in line 253 is not correct.

The sentence in line 307-309 should be interrupted at the middle, rather than a long sentence in the current version.

The sentence in line 358-360 begins with the phrase “As a result of …” and should be interrupted after the word “missing tip images”, instead of current version without stopping.

There are also many other places where long sentences like above could be modified or simplified. 

(2) The colored lines in the result pictures, e.g. Fig.10-12, have covered the appearance of the raw data and so it's not easy to judge which one is better. It would be better to use dotted lines instead of solid lines.

(3) It would be easier to understand in line 45-54, if the corresponding picture could be included.

(4) The hours in Table 2 could be estimated by computing the frames and fps, instead of “?” int the table.

(5) In table 4, the DLC with resnet 101 performs worse than with resnet 50 may because the number of trained data is not enough. The authors could perform pretraining of the resnet-101 on other similar tasks before training here. The performance should be better than the current version.

(6) There should be an extra conclusion section to summary the work, although there is an conclusion paragraph in the current version. In the conclusion section, the authors could give some future meaningful advice for the related researchers in the community, beyond the conclusion that DLC can be used here.

In summary, the topic of the work is meaningful, but the presentation should be revised carefully. 

Author Response

Thank you for taking the time to review this paper.

The paper mainly investigates the effect of using the models designed for pose estimation to estimate the speech articulator key-points without markers. This exploration is meaningful. But there are some problems with the manuscript.

(1) The writing is not so good and it's a little hard to read the manuscript. The main problem is that there are too many unnecessary long sentences. For example, the "-ing" phrase in line 260 is almost a complete sentence, which would be better to use a clause instead of the lengthy "-ing" phrase. The complex explanation in lines 260-263 has been deleted as it is not central to the topic of the paper.

The single sentence in line 261-263 includes a second clause in the first attributive clause, which is really not comfortable to read this long sentence. Deleted

The sentence in line 253 is not correct. Sentence deleted.

The sentence in line 307-309 should be interrupted at the middle, rather than a long sentence in the current version. Sentence has been broken in two parts.

The sentence in line 358-360 begins with the phrase “As a result of …” and should be interrupted after the word “missing tip images”, instead of current version without stopping. Comma added

There are also many other places where long sentences like above could be modified or simplified. Punctuation has been added in other places and long sentences have been simplified.

(2) The colored lines in the result pictures, e.g. Fig.10-12, have covered the appearance of the raw data and so it's not easy to judge which one is better. It would be better to use dotted lines instead of solid lines. We tried dotted lines but it made the lines themselves indistinct without fully revealing the underlying ultrasound.  When comparing the performance, the dark blue line is the hand label and it is most important that each algorithm is compared to this line.  We thought about having naked ultrasound frames but this would make the paper too long.  The raw data is available online on our github page. 

(3) It would be easier to understand in line 45-54, if the corresponding picture could be included. Lines 45-54 have been deleted due to a request from another reviewer to shorten the introduction.

(4) The hours in Table 2 could be estimated by computing the frames and fps, instead of “?” int the table. Training time is not related to frames and fps. It is related to the input image size, the number of training iterations and time required to update network weights at each iteration. We have asked the authors for an estimate of training time so that we can replace the “?” with values. As a consequence of their response we have replaced the ‘?’s with 2 hours in both cases.

(5) In table 4, the DLC with resnet 101 performs worse than with resnet 50 may because the number of trained data is not enough. The authors could perform pretraining of the resnet-101 on other similar tasks before training here. The performance should be better than the current version. We have reworked the final figure in Appendix B (RMSE scores vs iterations for each backbone). From that revised figure we can see that it is possible that Resnet101 has been slightly overtrained. However there is a level of variance in model performance depending on training iterations. Resnet101 ranges from 5.9-6.1 pixel RMSE while resnet50 ranges from 6.0-6.1. ResNet101 and all the other backbone networks are pretrained on human pose estimation data. There is however, no extra ultrasound tongue imaging data to pretrain Resnet101.

(6) There should be an extra conclusion section to summary the work, although there is an conclusion paragraph in the current version. In the conclusion section, the authors could give some future meaningful advice for the related researchers in the community, beyond the conclusion that DLC can be used here.

We have included suggestions for further work and published a github with extensive practical details and scripts to make using this open tool even easier. Another reviewer requested a reduction in the length of the discussion so have not added another section.

In summary, the topic of the work is meaningful, but the presentation should be revised carefully. 

Reviewer 3 Report

Contributions:

This study proposes using DeepLabCut with transfer learning for the pose estimation of speech articulators. My comments are given below:

  1. The authors pay great efforts to the conduction of experiments. However, the novelty is limited. The authors should address the novelty of this paper.
  2. In this paper, all neural networks been used can be found in the literature.
  3. The sub-grid lines should be removed in Figs.7-9,15, 16, 20, and 21.
  4. (Page 5) The contents in each block of Fig. 1 are too redundant. The contents can be addressed in the context.
  5. (Page 6) The caption of Fig. 2 is not well presented. Please make a table to introduce the label of each position.
  6. (Page 16) The title of each subplot should be removed.  
  7. (Page 15) Figures 7 and 8 are unclear. The performance has been presented in Table 4. Please remove these two figures.
  8. (Page 15) Please remove the titles. Figures 15, 16, 20, and 21 also have the same problem.
  9. (Page 26) The titles should be removed. In addition, the caption for this figure is missed.  

Author Response

Thank you for taking the time to review this paper.

This study proposes using DeepLabCut with transfer learning for the pose estimation of speech articulators. My comments are given below:

  1. The authors pay great efforts to the conduction of experiments. However, the novelty is limited. The authors should address the novelty of this paper. This paper uniquely demonstrates the equivalence of this fully automated pose estimation to hand labelling in the domain of ultrasound tongue imaging. Around 150 speech science laboratories worldwide  use other less effective manual, semi-automatic or automatic methods for ultrasound tongue contour estimation upon which they base their research.  We hope this paper provides evidence to support its use in this field. We have reworded the discussion section to clarify the novelty.
  2. In this paper, all neural networks been used can be found in the literature. Yes. The purpose of this paper is not to develop and test a new network but to evaluate the performance of freely available network training and inference software on a domain where edge detection algorithms are the state-of-the-art. For researchers to have confidence in adopting DeepLabCut as the preferred method for objectively annotating articulatory features they need evidence of its performance and benefits with respect to existing techniques.
  3. The sub-grid lines should be removed in Figs.7-9,15, 16, 20, and 21. We feel the horizontal grid lines help the reader assess the y-axis values for all of the speakers along the x-axis. For figure 9 we have removed all the gridlines as requested.  
  4. (Page 5) The contents in each block of Fig. 1 are too redundant. The contents can be addressed in the context. We have removed excess information from each block.
  5. (Page 6) The caption of Fig. 2 is not well presented. Please make a table to introduce the label of each position. A table has been added and the caption text description of each point removed.
  6. (Page 16) The title of each subplot should be removed.  The title acts as a legend for each subplot. If we remove the titles from each subplot in Figure 9 then it will be less easy for the reader to tell which method is associated with each.
  7. (Page 15) Figures 7 and 8 are unclear. The performance has been presented in Table 4. Please remove these two figures. The overall performance is given in the tables but it is relevant to see how the performance varies from speaker to speaker. For example if DeepLabCut performed more poorly on speakers TH DF and PB it might indicate that it only worked on ultrasound systems and probes that it had been trained on.  These figures demonstrate that this is not the case. Similar figures are used in Laporte & Menard 2018 to evaluate SLURP.
  8. (Page 15) Please remove the titles. Figures 15, 16, 20, and 21 also have the same problem. We have removed the titles as requested.
  9. (Page 26) The titles should be removed. In addition, the caption for this figure is missed.  We have removed the titles and added a caption. 

Reviewer 4 Report

This reviewer greatly appreciates the practical skills of the authors and the corresponding enthusiasm. However, the paper is too long, and a lot of information is provided without properly emphasizing the theoretical contribution of the authors. Sections 1 and 2 should be merged and the overall length should be considerably reduced. Table 1 adds unnecessary content, and it should be employed in the “Results” section to highlight contributions.

In all sections, the provided numerical information is presented in a somewhat chaotic manner and it greatly reduces the coherence of the paper. The practical aspects (values, chosen tools, etc.) should be briefly presented in the “Results” section. Consequently, the paper should better describe the theoretical state-of-the art, followed by clear contributions.

The authors decided to insert an overwhelming amount of data (parameter values, etc.) in all sections without providing information regarding the actual theoretical relevance or without performing comprehensive comparisons to standard methodologies/solutions. Section 4 seems forced – too many tools are briefly described without clear overall purpose – the authors should focus on the relevant information. The approach adds more confusion with respect to the above-mentioned emphasis on the contributions. What is the exact purpose of Section 4?

The overall impression is that the paper lacks proper theoretical content/substance, and it jumps from state-of-the-art information directly to “some” practical experiments.

Again, Section 6 is too long. What was the exact purpose of the paper? What are the exact contributions, and with respect to which standard approaches?

Other issues should be corrected, such as missing commas; example: "...the tongue but is usually...". Also, do not use the word "but" (too informal). More examples can easily be provided.

Author Response

Thank you for taking the time to review this paper.

This reviewer greatly appreciates the practical skills of the authors and the corresponding enthusiasm. However, the paper is too long, and a lot of information is provided without properly emphasizing the theoretical contribution of the authors. Sections 1 and 2 should be merged and the overall length should be considerably reduced. Table 1 adds unnecessary content, and it should be employed in the “Results” section to highlight contributions.

We have merged Sections 1 and 2 as requested. We have deleted some paragraphs to shorten the paper.  Table 1 summarises the performance of state-of-the-art algorithms reported in the literature that apply to automatic tongue contour estimation. We use these reported results as a basis for selecting algorithms to compare with pose-estimation. We have retitled the section heading and reworded the start and end of the section to emphasise this point.

In all sections, the provided numerical information is presented in a somewhat chaotic manner and it greatly reduces the coherence of the paper. The practical aspects (values, chosen tools, etc.) should be briefly presented in the “Results” section. Consequently, the paper should better describe the theoretical state-of-the art, followed by clear contributions.

We have tried to reduce the chaos by moving discussion of keypoint labelling to the “materials and methods” section. We feel that practical aspects (values, chosen tools…) should be presented in the “Materials and methods” section. The “Results” section should contain only the results. In tables 5 and 7 we have deleted RMSE scores apart from test set with P>0.6 in an effort to simplify the numerical data.

The authors decided to insert an overwhelming amount of data (parameter values, etc.) in all sections without providing information regarding the actual theoretical relevance or without performing comprehensive comparisons to standard methodologies/solutions.

The paper describes three separate experiments to evaluate the performance of DeepLabCut in the field of speech Science.  These are: Experiment 1 Evaluate the accuracy of ultrasound tongue contour estimation with respect to hand labelling and three state-of-the-art techniques. The introduction section and table 1 explains the reasons for selecting these state-of-the-art methods. Comparison is made with these state-of-the-art methodologies in table 4. Experiment 2. Evaluate the accuracy of pose estimated keypoints in comparison with physical sensors glued to the tongue. Experiment 3: Evaluate the accuracy of lip contour estimation with respect to hand labelling. Three related experiments all require description of materials and methods.  This does unfortunately lead to a lot of data. This 3rd experiment is necessary to show that pose estimation can be applied successfully to other speech science data streams. We omitted our work on MRI data partly because it would have made the paper too long and we had already made the point with these three experiments.

Section 4 seems forced – too many tools are briefly described without clear overall purpose – the authors should focus on the relevant information. The approach adds more confusion with respect to the above-mentioned emphasis on the contributions. What is the exact purpose of Section 4?

We have changed the subheading titles in the “materials and methods” section (section 4) in order to make the purpose of each section clearer. Section 4 describes in detail the settings for the three state-of-the-art ultrasound tongue contour estimation techniques (SLURP, MTracker and DeepEdge) selected for comparison with DeepLabCut. They were selected based on a thorough review of reported performances and availability of code in section 1. Section 4 also describes the methodologies for the comparison of tongue contours, for comparison of DLC with Electromagnetic articulography and for the evaluation of lip contours. This section additionally describes the datasets used for these comparisons. 

The overall impression is that the paper lacks proper theoretical content/substance, and it jumps from state-of-the-art information directly to “some” practical experiments.

While not based on a novel theory or novel network architectures we believe this work constitutes original research and reports scientifically sound experiments. It provides a substantial amount of new information about the performance of transfer learning and pose estimation for the benefit of researchers in the field of speech science where this approach is virtually unknown. We show for the first time that the technique performs as well as human labellers in this broad domain. This result has not demonstrated in this domain by any other technique.  At least 150 speech science laboratories would benefit from using this approach rather than existing state-of-the-art techniques and this paper, we hope, will give them the confidence to do so.

Again, Section 6 is too long. What was the exact purpose of the paper? What are the exact contributions, and with respect to which standard approaches?

Another reviewer has requested that we extend this section. We have rewritten it to try to make the purpose of the paper and the novel contributions with respect to the state-of-the-art more cogent.

Other issues should be corrected, such as missing commas; example: "...the tongue but is usually...". Also, do not use the word "but" (too informal). More examples can easily be provided. This sentence and surrounding sentences have been removed. Missing commas have been added elsewhere in the paper.  Long sentences have been broken up into shorter sentences.

Round 2

Reviewer 1 Report

I thank the authors for addressing my comments. 

Regarding table 4, I understand the reasons of the authors. However, I believe that interpreting the table would be much easier for readers if the authors include a discussion similar to what reported in their reply.

Author Response

Thank you again for your time. We believe your review has helped improve the paper. We have revised comments in section 4 relating to Table 4. We have added to the Discussion section with remarks made in our previous reply to you. Where relevant, we have also amended the description of the Edge Detection algorithms in Section 3 to avoid duplication of comments.

Reviewer 3 Report

The quality of this paper has been improved. I think this paper can be accepted for publication.

Author Response

Thank you again for your time. We agree that your review has helped improve the paper.

Reviewer 4 Report

The quality of the presentation has improved. This reviewer's concerns were properly addressed.

Minor checks should be made to eliminate remaining formatting issues, such as extra space, wrong indentation, etc.

Author Response

Thank you again for your time. Your review has help improve the paper.  We have corrected various formatting errors as well as some typos in a further revision.